# Impact of the COVID-19 Pandemic on the Female Sexual Function Index and Female Behavioral Changes: A Cross-Sectional Survey Study in Thailand

**DOI:** 10.3390/ijerph192315565

**Published:** 2022-11-23

**Authors:** Udomsak Narkkul, Jun Jiet Ng, Apisith Saraluck

**Affiliations:** 1Department of Medical Science, School of Medicine, Walailak University, Nakhon Si Thammarat 80160, Thailand; 2Research Center in Tropical Pathobiology, Walailak University, Nakhon Si Thammarat 80160, Thailand; 3Department of Obstetric and Gynaecology, Kuala Lumpur Hospital, Jalan Pahang, Kuala Lumpur 50586, Malaysia; 4Department of Clinical Medical Science, School of Medicine, Walailak University, Nakhon Si Thammarat 80160, Thailand

**Keywords:** COVID-19 pandemic, female sexual function index, female behavioral changes, Thailand

## Abstract

Sexual health alterations are associated with disasters. Consequently, the COVID-19 pandemic may affect female sexual function. This study aimed to determine the COVID-19 pandemic effect on female sexual function and to know the risk of female sexual dysfunction. This online, cross-sectional, observational research was conducted during the pandemic period. A logistic regression model was used to investigate the associations between outcomes and potential risk factors. In total, 432 sexually active women participating in the region affected by the COVID-19 pandemic were analyzed. The overall findings of our study are that 60 percent of females were at risk for female sexual dysfunction. The average FSFI score was 21.27 ± 7.17. Comparing female sexual behavior before and during the COVID-19 pandemic reveals a significant decrease in the frequency of having sex per week, foreplay duration, and coital duration. In the multivariate analysis, the factors associated with the development of RFSD are age greater than 45 years (adjusted odds ratios (AOR) 15.09, 95% confidence interval (CI) 3.67–62.07), body mass index (BMI) greater than 25 (AOR 3.26, 95%CI 1.23–8.67), jobs as a healthcare provider (AOR 8.45, 95%CI 3.66–19.53), previous COVID-19 infection within the previous three months (AOR 36.81, 95%CI 10.93–123.98), and screened-positive anxiety (AOR 13.07, 95%CI 4.75–35.94). COVID-19 influences female sexual behavior and may increase the risk of sexual dysfunction in women. Concern for the effects of female sexual quality of life in high-risk individuals is essential.

## 1. Introduction

The coronavirus disease 2019 (COVID-19), an infectious illness brought on by the SARS-CoV2 virus, was considered a pandemic by the World Health Organization (WHO), and several states implemented harsh regulations, including quarantine, border closures, and transit restrictions [1]. When in close contact with an infected individual, exposed through respiratory droplets or aerosols, COVID-19 transmission happens quickly via the nose and mouth [2]. COVID-19 has profoundly disrupted social relationships and health services that are fundamental to sexual and reproductive health [3]. A wide range of essential sexual and reproductive health services was stopped or reoriented because of the pandemic [4]. Sexual health is influenced by a variety of factors, including physical, emotional, social, and psychological [5]. Sexual health problems or changes in sexual behavior can be caused by a person’s feelings, stress, the environment, and the social situation. Numerous reports indicate that disasters around the world, such as earthquakes, hurricanes, or outbreaks of diseases, influence female sexual behavior and happiness [6]. Naturally, tension, anxiety, and concern about contact and infection may increase in COVID-19 pandemic zones. The association between disaster, i.e., earthquakes or hurricane, and female sexual function has been the focus of several research studies [7,8], and in recent years, a few papers have revealed how COVID-19 impacts female sexual dysfunction.

Female sexual behavior and female sexual dysfunction are still unclear in relation to the COVID-19 epidemic. Some studies that were done in different countries showed that people’s sexual behavior had changed. According to NBC News, a study of more than 9000 people found that more than 47% said the COVID-19 infection had made changes in their lives for the worse [9]. Additionally, a study in China found that both young men and women were less sexually active [10]. However, a survey of how lockdown situations affect sexual activity in Bangladesh, India, and Nepal showed that 3.3% of participants increased having sexual relations [11]. The different places are different in culture and lifestyle, which also affects the physical, emotional, and sexual behavior and function of the people. However, some research showed that anxiety and depression may have an effect on the risk of female sexual dysfunction in the context of the COVID-19 epidemic. Additionally, some studies indicate that during the COVID-19 pandemic, being a female healthcare provider increases the risk of female sexual dysfunction and decrease the frequency of sexual activity [12,13]. 

In mid-2021, the Department of Disease Control, Ministry of Health Thailand, stated that the number of COVID-19-infected individuals had exceeded 1.5 million and was continuing to increase rapidly at 10,000 per day [14]. Nakhon Si Thammarat, one of the largest cities in Thailand’s southern region, has seen numerous outbreaks and has the greatest number of patients with COVID-19 infection across the country. As a result of the pandemic in Thailand, healthcare professionals in pandemic-affected areas are under greater pressure and work longer hours [15], which may have an impact on sexual behavior and lead to sexual dysfunction. We hypothesize that this pandemic may result in an increase in sexual behavior alterations and female sexual dysfunction. The purpose of this study is to determine the effects of the COVID-19 pandemic on female sexual function and to know the risks of female sexual dysfunction. The results of this study can be utilized to provide advice for the management of female sexual dysfunction and to provide patient counseling.

## 2. Materials and Methods

### 2.1. Study Population and Sample Size

This cross-sectional, observational research was conducted from 1 December 2021 to 1 March 2022. We collected data in Nakhon Si Thammarat province (the largest province in the southern region of Thailand). The study enrolled sexually active women of reproductive age over the age of 18 who had been living with their partners for more than six months and who lived or worked in Nakhon Si Thammarat during the outbreak period, at least since April 2021. The main exclusion criteria were the presence of sexual dysfunctions, inactivity, age under eighteen, use of libido-suppressing medication in the preceding three months, and personality disorders or other mental illnesses, including depression and marital conflict. Women included in our study were invited to participate in a voluntary, anonymous online survey. Prior to the survey’s initiation, each participant was given an informed consent form, and completion of the survey constituted informed consent to participate in the study. This study was approved by the Human Research Ethics Committee of Walailak University (WU-EC-OT-2-306-64).

The sample size was determined using the single population proportion formula in the n4Studies application [16]. Using the data from a previous study, the proportion of sexual dysfunction was 50% (*p* = 0.5), with a 95% confidence interval (CI; z = 1.96) and a 5% margin of error (d = 0.05) [12]. The sample size was calculated to be 385. We assumed that the final sample size would be reduced by 10% owing to incomplete data and non-response rate; therefore, the final sample size in this study was 432.

### 2.2. Data Collection and Survey Instruments

A convenient sampling technique was applied to select the study participants. A self-reported questionnaire was designed using the Google survey tool (Google Forms; Google, Mountain View, CA, USA). The generated link was converted into a quick response (QR) code with an explanation and online consent, shared with the online platform of government offices in every district of Nakhon Si Thammarat Province, and labels were also attached to the QR code to invite individuals who entered in accordance with the inclusion criteria at government offices and community offices. The QR code was shared afterward with the public on social media, such as via Facebook, as well as via Line and Twitter accounts.

All participants completed online surveys that included questions regarding their demographic features, prior medical history, history of COVID-19 infection in the last 3 months, sexual behavior changes, and the Female Sexual Function Index (FSFI) [17]. The FSFI is a standardized questionnaire comprised of 19 questions, each assessed on a Likert scale from 0/1 to 5, for a maximum total score of 36, with six domains scoring six points each: desire, arousal, lubrication, orgasm, satisfaction, and pain [17]. The presence of risk for female sexual dysfunction (RFSD) was defined as an FSFI score of less than 26.55 [17,18]. We use the Thai version of the FSFI, which Peeyananjarassri et al. translated and validated [19]. Peeyananjarassri et al. only did the translation and back-translation, evaluated comprehension and wording, and calculated the reliability coefficient. The reliability coefficient of the FSFI Thai version was 0.9 [19]. The survey included questions about sexual behavior changes between pre- and post-pandemic periods in regards to sexual frequency, masturbation frequency, duration of foreplay, and duration of sexual intercourse. The validated Patient Health Questionnaire for Depression and Anxiety with four items was used to assess anxiety and depression symptoms (PHQ-4) [13]. The PHQ-4 is a validated screening test for anxiety and depression that consists of two questions about depression (PHQ-2) and two questions about generalized anxiety disorder (GAD-2). Each question is scored on a Likert scale ranging from 0 to 3, for a total of six possible points for each subscore, and scores of 3 for anxiety or depression were considered high on their respective scales [20].

### 2.3. Statistical Analysis

Data were entered into an Excel database and subsequently double-checked for validation before analytical processing. All data analyses were performed using IBM SPSS Statistics for Windows, Version 23.0 (SPSS, Chicago, IL, USA). The survey data were analyzed for both descriptive and inferential statistics. Continuous variables, including age, weight, height, BMI, desire, arousal, lubrication, orgasm, satisfaction, pain, and total scores, were described using mean and standard deviation (SD). Independent categorical variables, including demographic and socioeconomic data (i.e., age, BMI, education level, occupation, type of work, religion, congenital disease, COVID-19 infection, family’s COVID-19 infection, smoking, alcohol consumption, anxiety, depression, and female sexual dysfunction) and information on sexual behavior between pre-COVID-19 and intra-COVID-19 were described using frequency and expressed as percentages. Univariate analysis was used to examine the crude odds ratio (OR) of the binary outcome variable for each independent variable. All variables in the univariate analysis were subjected to multivariable analysis to adjust for possible confounders by calculating adjusted odds ratios (AOR) with 95% CIs. Statistical significance was set at *p* < 0.05. A logistic regression model was used to investigate the associations between outcomes and potential risk factors.

## 3. Results

### 3.1. Sociodemographic Characteristics

In total, 432 sexually active women that participated in the questionnaire in the region affected by the COVID-19 pandemic were analyzed. The mean age of study participants was 32.87 years with a standard deviation of 10.81 years and a range from 18 to 60 years. Most participants had a normal BMI (91.9%) and no underlying medical condition (89.3%). The majority of participants reported holding a bachelor’s degree or higher (98.1%). Over one-third (35.8%) of respondents were healthcare providers, and the majority (81.2%) were working full time. Most of the participants (86.3%) were Buddhists. Approximately one-third (34.9%) of respondents had a history of COVID-19 infection positivity. During the pandemic, 199 women (46.1%) screened positive for anxiety, and 48 women (11.1%) screened positive for depression (Table 1).

### 3.2. Sexual Behavior between Pre-COVID-19 and Intra-COVID-19

Comparing female sexual behavior before and during the COVID-19 pandemic reveals a significant decrease in the frequency of having sex once per week. Prior to the pandemic, the majority of women experienced sexual activity 5–10 times per week, whereas during the pandemic, the majority of women engaged in sexual activity 3–4 times per week. Before the pandemic, approximately 60% of women had foreplay ranging 11–15 min, while during the pandemic, the majority of them (60.4%) had foreplay ranging 5–10 min. Furthermore, prior to the pandemic, 46.7 percent of women had coital duration times of 3–5 min and 35.4 percent had 5–10 min, whereas during the pandemic, 45.3 percent of women had coital duration times of only 1–2 min and more than 10% of females had coital duration times of more than 10 min. There was a significant difference in masturbation frequency between before and during the COVID-19 pandemic. Before the outbreak, 50% of women reported masturbation 1–2 times per week, and 32.6 percent reported 3–5 times per week; after the pandemic, 58 percent reported 1–2 times per week, and 23.6 percent reported 3–5 times per week (Table 2).

### 3.3. Female Sexual Dysfunction

Table 3 shows that the incidence of women with the presence of risk of female sexual dysfunction (RFSD) during the pandemic was 60.8% (263/432). 

### 3.4. Domain

Six domains of the FSFI’s average scores were shown in the Table 4. This study demonstrated that the average scores for desire, arousal, lubrication, orgasm, satisfaction, and pain were 3.48, 3.22, 3.39, 3.32, 3.74, and 4.13, respectively. The average FSFI score was 21.27.

### 3.5. Factors Associated with Female Sexual Dysfunction

In the univariate analysis of factors related to the development of RFSD, age greater than 45 years (OR = 18.16), occupation as a healthcare provider (OR = 14.08), previous COVID-19 infection in the past 3 months (OR = 52.27), partner’s previous COVID-19 infection in the past 3 months (OR = 1.54), screened-positive anxiety (OR = 62.58), and screened positive depression (OR = 5.16) were associated with the development of RFSD. For the multivariate analysis, the factors associated with the development of RFSD were age greater than 45 years (AOR = 15.09), BMI greater than 25 (AOR = 3.26), jobs as a healthcare provider (AOR = 8.45), previous COVID-19 infection within the previous three months (AOR = 36.81), and screened-positive anxiety (AOR = 13.07) (Table 5).

## 4. Discussion

Sexual function and intimacy are not only important for species propagation but are an important component of quality of life. The COVID-19 pandemic has greatly impacted every aspect of human life, including sexual function. According to the Diagnostic and Statistical Manual of Mental Disorders (DSM-5), female sexual dysfunction (FSD) is defined as “any sexual complaint or problem resulting from disorders of desire, arousal, orgasm or sexual pain that causes marked distress or interpersonal difficulty” [21]. The Female Sexual Function Index (FSFI) is a standard multidimensional tool for assessing female sexual function. The 19-item measure assesses sexual function over the past four weeks and provides scores in six domains, including sexual desire, arousal, lubrication, orgasm, satisfaction, and pain [17]. The optimal cut score for identifying women with sexual dysfunction is an FSFI total score of less than 26.55. Different studies and literature have quoted the prevalence of FSD between 30% and 60% [22]. In our study, the prevalence was 60%, which is on the upper limit of the normal range. Some previous research in each region of the world, such as Turkey and the United States, indicate a significantly increased incidence of female sexual dysfunction during pandemics [5,6]. This is a serious problem that should be looked into because FSD can affect emotional well-being leading to relationship strain, break-ups, divorce, family breakdown, and even suicide. The causes of FSD are multifactorial and can be classified into physical and psychological. COVID-19 can cause FSD both physically and psychologically. Almost half of the participants in our study experienced anxiety, depression, panic, and post-traumatic stress disorder during the COVID-19 pandemic due to the loss of loved ones, financial instability, unemployment, and the uncertainty of the future. This explained the findings of a higher prevalence of FSD among participants that screened positive in the Patient Health Questionnaire for Depression and Anxiety (PHQ-4). Activity restriction, immobility, and reduced entertainment and recreational activities have impacted sexual well-being and functioning as well. Some of the participants that had COVID-19 infection in the last 3 months experienced long-term effects of COVID-19 such as chronic fatigue syndrome, chronic cough, headache, palpitation, and insomnia, which reduced their interest in sex and their sexual functioning [23]. As the natural illness COVID-19 spreads by person-to-person contact and is easily transmissible, there is a heightened concern for the overall community. Due to the fact that sexual activity is an action that is associated with human interaction, especially during the COVID-19 pandemic era when the disease is easily transmissible, it is quite likely that sexual behavior will be affected. Due to stress, social interactions between couples, the job, and society may have an impact on sexual behavior.

Sexual behavior is the manner in which humans express their sexuality and is not only limited to penetrative sex [23]. Masturbation, foreplay, and oral sex are examples of different sexual behaviors. COVID-19 is an airborne disease that spreads through person-to-person close contact, and it influences female sexual behavior as well. Sexual behavior is accessed through the frequency of sex, coital duration, foreplay duration, and frequency of masturbation. In our study, there was a decrease in the frequency of sex per week, as well as the coital duration and foreplay duration, especially among healthcare professionals. Long working hours, high levels of chronic stress, and occupational burnout among healthcare providers explained the findings of our study. There was an increase in the rate of masturbation during the COVID-19 pandemic. Masturbation is preferred over sex because of the fear of contracting COVID-19 through close contact. In addition, couples might be separated and have less chance to meet up due to travel restrictions and public health measure implementation. It is recognized that a stressful lifestyle affects female sexual desire and frequency of sexual activity. The research of Yuksel and OzgOr revealed an increase in the number of sexual encounters [5]. They stated that during the epidemic, families spend more time together at home. However, Guzel and DonDu also found a decline in sexual activity among healthcare professionals during the COVID-19 pandemic [12]. Studies in the literature have comparable outcomes as ours. According to Hamilton and Meston, high levels of chronic stress diminish sexual desire [24]. Similarly, Liu et al. discovered a decline in sexual activity following an earthquake [7].

To the best of our knowledge, our study highlights the identification of the risk factors for female sexual dysfunction. Occupation as a healthcare practitioner, prior COVID-19 infection within three months, a positive screening for anxiety, age more than 45 years, and BMI greater than 25 were identified as risk factors from multivariate analysis. Due to the COVID-19 epidemic era, it is known that healthcare providers have more contact with patients who are at risk for or are positive for COVID-19. Our study reports, as did a previous study conducted in Turkey specifically on healthcare providers, that the RFSD has increased significantly in both male and female healthcare providers [23]. In another study from the epicenter of Brazil, 37% of healthcare professionals reported a worsening of libido and sexual dysfunction because of various factors, such as a loss of nightlife and isolation from one’s spouse [25]. Sexual dysfunction is usually linked to psychological issues. During the COVID-19 epidemic, healthcare practitioners have a greater burden and spend more time working, which may have an impact on their emotional and psychological well-being. The latest evidence on the psychological effect of COVID-19 on health professionals discovered that 29.8% of them experience stress, 24.1% experience anxiety, and 13.5% experience depression [26]. It has been shown that the psychological impacts of this worldwide pandemic on healthcare personnel have an impact on their sexual life. Interestingly, a previous survey on the sexual attitudes among healthcare workers during the COVID-19 outbreak found that 50% of healthcare workers were concerned about transmitting COVID-19 through sexual intercourse after virus RNA was discovered in sperm [27]. The lack of certainty in this scenario may also cause anxiety among healthcare practitioners and increase the risk of female sexual dysfunction. Moreover, a recent paper on the factors affecting sexual dysfunction in healthcare workers found that female gender is a risk factor for sexual dysfunction, and since all of the participants in our study were female, which may suggest that women who work as healthcare professionals are at risk for FSD in our study [12]. 

Our data indicate that individuals who have previously tested positive for COVID-19 are at risk for RSFD. Multiple variables may account for the reduction in sexual function in COVID-19 patients. In difficult times, it is a truth that one’s way of life must be altered. It is considered that societal constraints and uncertainty about the future have an impact on people’s lifestyle and sexual function. A previous research study, which was a prospective study limited to COVID-19-positive patients, demonstrated a significant increase in the incidence of RSFD in females following COVID-19 positivity [28]. According to a recent study, COVID-19 seriously impacts men’s sexual health and erectile function in particular ways through physical health, mental health, and medical access processes [29]. On the other hand, our study shows that COVID-19 positivity in women is a known risk factor for female sexual dysfunction. The major ways that COVID-19 spreads are through respiratory droplets and direct contact between people. Although it is unknown if COVID-19 is sexually transmitted, intimate contact between partners because of sexual activity may have an impact on how the virus spreads. Based on what the author thinks, if a man has erectile dysfunction because of COVID-19, a woman may also have sexual dysfunction, either because of the condition itself or because of how it affects her partner. Moreover, social isolation during COVID-19 positivity may raise the probability of female sexual dysfunction. Acute stress, which manifests itself throughout the period of social isolation, may affect patients’ sexual performance issues. It has been established that stress from social isolation is the fundamental cause of sexual dysfunction [30]. 

The COVID-19 pandemic has caused sleep disturbance, anxiety, stress, and depressed behavior as mental health issues. The development of psychiatric problems has been documented both during and after COVID-19 infection [31]. The mechanism involved in the increased mental issues in COVID-19 patients is the cytokine storm or the cytokine release syndrome, which was demonstrated by elevated plasma levels of pro-inflammatory cytokines [32]. Depression and anxiety are the most common psychological problems that affect sexual dysfunction in the aftermath of the COVID-19 pandemic [33]. Extensive research has been conducted on the direct association between female sexual dysfunction during the COVID-19 pandemic, anxiety, and stress. According to a recent study conducted in Egypt, which found that anxiety was a predictor of sexual relationship stress in females, our investigation showed that a positive result on an anxiety screening is a risk factor for female sexual dysfunction [34]. Similarly, Hamilton and Meston observed that female sexual dysfunction was caused by high amounts of chronic stress brought on by COVID-19, which decreased sexual desire [24]. Numerous pharmacologic therapies for anxiety include FSD as a side effect, but anxiety itself is a risk factor for RFSD, even in the absence of treatment. In one trial of placebo-treated individuals with generalized anxiety disorder, 46% of women developed sexual dysfunction [12]. Studies have been done on the correlation between female sexual dysfunction and the prevalence of depressive symptoms. An inversely proportionate association between depression and FSFI was discovered in a prior study that was conducted in Polish women, but the correlation’s strength was low [35]. On the other hand, Cohen et al. found that persons with depressive conditions had a higher rate of sexual dysfunction [36]. Ilgen et al. demonstrated that during the pandemic, anxiety and sadness levels increased [37]. However, our study indicated that although screening for depression tends to raise the risk of female sexual dysfunction, the multivariate analysis did not find this to be significant. Our data suggest that age more than 45 and BMI greater than 25 are risk factors for female sexual dysfunction. However, there are few research studies that focus on the impact of age and BMI on female sexual dysfunction. A previous study discovered that during COVID-19, a woman’s future reproductive potential in adolescence may be significantly impacted due to an unintended consequence of a reduction in services and the inaccessibility of sexual and reproductive healthcare but did not include FSD risk [38]. Furthermore, females over the age of 45 are more likely to have dyspareunia, which can reduce domain pain in the FSFI score, while females under the age of 45 are less likely to have dyspareunia [39]. Our analysis indicated that a BMI more than 25 is also associated with an increased risk of female sexual dysfunction; interestingly, reports on males also show that obesity is associated with an increased risk of erectile dysfunction [40].

This is the first study to evaluate sexual function in Southeast Asian women during the COVID-19 outbreak. Early research focusing on the correlation between female sexual dysfunction and the COVID-19 outbreak were undertaken in the United States of America and Europe and may not be generalizable because each country’s medical and psychological reaction to the pandemic has been unique. We gathered information from a large number of study participants using an online survey tool that gave them the flexibility to respond to questions on sexuality. Furthermore, the strength of our research is the reporting of female behavioral changes over the COVID-19 era and the analysis of risk factors that could impact female sexual dysfunction, which is important to be aware of in terms of clinical implications and health promotion in high-risk patients.

There are several limitations to this research. Although we tried to spread our questionnaire across provinces, areas, and occupations, this study was an online survey that was promoted through various online media channels and conducted during the peak of the COVID-19 pandemic. A large portion of survey respondents might have been health professionals, since health professionals are aware of health-related issues. It is also possible that there were some biases because online surveys are more likely to be filled out by those who use social media and computers or laptops at work, such as health professionals, for example. The respondents could respond during office hours, so this might have encouraged more of those who had access to the internet during office hours. It is probable that, for example, other occupations and those who do not use computers or social media at work did not receive the invitation that came via social media. Thus, they received the invitation and instructions to fill out the survey online but might not have done so during their leisure time. This study is cross-sectional, and hence some of the findings may be recall biased when compared to previous COVID-19 circumstances. Furthermore, the study only looked at female sexual behavior; however, future research may look at how male sexual behavior may alter female sexual attitudes in a pandemic.

## 5. Conclusions

In conclusion, COVID-19 pandemic has a major impact on female sexual function and behavior. Sexual function and sexual quality of life during the COVID-19 pandemic declined dramatically compared to before the COVID-19 pandemic. The prevalence of female sexual dysfunction is high in the pandemic area, and this should be taken seriously, as female sexual dysfunction may lead to devastating consequences. Preventive measures, screening, and appropriate treatments are the way forward. Women with high risk for FSD should be screened for female sexual dysfunction, as there is an association between mental health, occupation, socioeconomic status, and sexual health. The information from this study can be used for patient counseling and to guide the management of female sexual dysfunction. Additionally, our study provides valuable data for meta-analysis and systemic review, combining studies from all around the world. 

## Figures and Tables

**Table 1 ijerph-19-15565-t001:** Demographic characteristics.

Characteristics		Total (*n* = 432)	
		Number	Percentage
Age (Mean ± SD)		32.87 ± 10.81	
Weight (Mean ± SD)		59.48 ± 8.63	
Height (Mean ± SD)		164.55 ± 5.75	
BMI (Mean ± SD)		21.94 ± 2.75	
Age group			
	<45	364	84.26
	≥45	68	15.74
BMI group			
	<25	397	91.90
	≥25	35	8.10
Education level			
	High school	8	1.85
	Bachelor’s degree	362	83.8
	Postgraduate	62	14.35
Occupation			
	Not related to health	277	64.12
	Related to health	155	35.88
Type of work			
	None	68	15.74
	Part time	14	3.24
	Full time	350	81.02
Religion			
	Buddhism	372	86.31
	Christianity	6	1.39
	Islam	53	12.3
Underlying disease			
	No	386	89.35
	Yes	46	10.65
Previous COVID-19 infection in past three months			
	No	281	65.05
	Yes	151	34.95
Partner had COVID-19 Infection in past three months			
	No	235	54.4
	Yes	197	45.6
Smoking status			
	No	410	94.91
	Yes	22	5.09
Alcohol consumption			
	No	369	85.42
	Yes	63	14.58
Screened Anxiety Positive by PHQ-2			
	No	233	53.94
	Yes	199	46.06
Screened Depression Positive by PHQ-21			
	No	384	88.89
	Yes	48	11.11

**Table 2 ijerph-19-15565-t002:** Sexual behavior between pre-COVID-19 and intra-COVID-19.

Characteristics		Sexual Behavior (*n* = 432)				*p*-Value
		Pre-COVID-19		Intra-COVID-19		
		Number	Percentage	Number	Percentage	
Average frequency of having sex during 1 week						<0.001
	0 times	19	4.40	23	5.32	
	1–2 times	60	13.89	121	28.01	
	3–4 times	90	20.83	176	40.74	
	5–6 times	121	28.01	99	22.92	
	7–10 times	121	28.01	12	2.78	
	>10 times	21	4.86	1	0.23	
Average frequency of masturbation during 1 week						0.004
	0 times	57	13.19	70	16.2	
	1–2 times	217	50.23	252	58.33	
	3–5 times	141	32.64	102	23.61	
	>5 times	17	3.94	8	1.85	
Foreplay duration						<0.001
	<5 min	33	7.64	145	33.56	
	5–10 min	109	25.23	261	60.42	
	11–15 min	250	57.87	19	4.4	
	>20 min	40	9.26	7	1.62	
Coitus duration						<0.001
	<1 min	7	1.62	26	6.02	
	1–2 min	25	5.79	196	45.37	
	3–5 min	202	46.76	137	31.71	
	5–10 min	153	35.42	28	6.48	
	>10 min	45	10.42	45	10.42	

**Table 3 ijerph-19-15565-t003:** Female sexual dysfunction (*n* = 432).

Characteristics		Female Sexual Dysfunction (*n* = 432)		*p*-Value *
		No (*n* = 169)	Yes (*n* = 263)	
Age (Mean ± SD) **		29.43 ± 6.2	35.08 ± 12.45	<0.001
Weight (Mean ± SD) **		59.8 ± 9.00	59.27 ± 8.39	0.533
Height (Mean ± SD) **		164.82 ± 5.74	164.37 ± 5.76	0.434
BMI (Mean ± SD) **		21.99 ± 2.99	21.90 ± 2.59	0.744
Age group				<0.001
	<45	166 (98.22)	198 (75.29)	
	≥45	3 (1.78)	65 (24.71)	
BMI group				0.541
	<25	157 (92.90)	240 (91.25)	
	≥25	12 (7.10)	23 (8.75)	
Education level				<0.001
	High school	4 (2.37)	4 (1.52)	
	Bachelor’s degree	157 (92.90)	205 (77.95)	
	Postgraduate	8 (4.73)	54 (20.53)	
Occupation				<0.001
	Not related to health	156 (92.31)	121 (46.01)	
	Related to health	13 (7.69)	142 (53.99)	
Type of work				0.221
	None	33 (19.53)	35 (13.31)	
	Part time	5 (2.96)	9 (3.42)	
	Full time	131 (77.51)	219 (83.27)	
Religion				0.690
	Buddhism	148 (88.1)	224 (85.17)	
	Christianity	2 (1.19)	4 (1.52)	
	Islam	18 (10.71)	35 (13.31)	
Underlying disease				0.011
	No	159 (94.08)	227 (86.31)	
	Yes	10 (5.92)	36 (13.69)	
Previous COVID-19 infection in past three months				<0.001
	No	165 (97.63)	116 (44.11)	
	Yes	4 (2.37)	147 (55.89)	
Partner had COVID-19 infection in past three months				0.028
	No	103 (60.95)	132 (50.19)	
	Yes	66 (39.05)	131 (49.81)	
Smoking status				0.128
	No	157 (92.9)	253 (96.2)	
	Yes	12 (7.1)	10 (3.8)	
Alcohol consumption				0.857
	No	145 (85.8)	224 (85.17)	
	Yes	24 (14.2)	39 (14.83)	
Anxiety				<0.001
	No	162 (95.86)	71 (27.00)	
	Yes	7 (4.14)	192 (73.00)	
Depression				<0.001
	No	163 (96.45)	221 (84.03)	
	Yes	6 (3.55)	42 (15.97)	

* *p* < 0.05, significantly different between groups using the chi-squared test. ** *p* < 0.05, significantly different between groups using the independent *t*-test.

**Table 4 ijerph-19-15565-t004:** Domain.

Characteristics		Total (*n* = 432)	
		Number	Percentage
Desire (Mean ± SD)		3.48 ± 1.16	
Arousal (Mean ± SD)		3.22 ± 1.39	
Lubrication (Mean ± SD)		3.39 ± 1.47	
Orgasm (Mean ± SD)		3.32 ± 1.4	
Satisfaction (Mean ± SD)		3.74 ± 1.24	
Pain (Mean ± SD)		4.13 ± 1.43	
Total score (Mean ± SD)		21.27 ± 7.17	
Risk of female sexual dysfunction			
	No	169	39.10
	Yes	236	60.90

**Table 5 ijerph-19-15565-t005:** Univariate and multivariate logistic regression analysis of factors influencing female sexual dysfunction (*n* = 432).

Characteristics		N ^1^	Female Sexual Dysfunction ^2^	OR ^3^ (95%CI)	AOR ^4^ (95%CI)
Age group					
	<45	364 (84.26)	198 (54.40)		
	≥45	68 (15.74)	65 (95.59)	18.16 (5.60–58.85)	15.09 (3.67–62.07)
BMI group					
	<25	396 (91.67)	240 (60.45)		
	≥25	36 (8.33)	23 (65.71)	1.25 (0.60–2.59)	3.26 (1.23–8.67)
Occupation					
	Not related to health	277 (64.12)	121 (43.68)		
	Related to health	155 (35.88)	142 (91.61)	14.08 (7.60–26.06)	8.45 (3.66–19.53)
COVID-19 infection					
	No	281 (65.05)	116 (41.28)		
	Yes	151 (34.95)	147 (97.35)	52.27 (18.82–145.14)	36.81 (10.93–123.98)
Family’s COVID-19 Infection					
	No	235 (54.4)	132 (56.17)		
	Yes	197 (45.6)	131 (66.5)	1.54 (1.04–2.29)	0.61 (0.28–1.31)
Smoking status					
	No	410 (94.91)	253 (61.71)		
	Yes	22 (5.09)	10 (45.45)	0.51 (0.21–1.22)	0.74 (0.17–3.15)
Alcohol consumption					
	No	369 (85.42)	224 (60.7)		
	Yes	63 (14.58)	39 (61.9)	1.05 (0.60–1.82)	1.72 (0.68–4.30)
Anxiety					
	No	233 (53.94)	71 (30.47)		
	Yes	199 (46.06)	192 (96.48)	62.58 (28.00–139.85)	13.07 (4.75–35.94)
Depression					
	No	384 (88.89)	221 (57.55)		
	Yes	48 (11.11)	42 (87.5)	5.16 (2.14–12.43)	0.85 (0.25–2.90)

^1^ Total sample of 432. ^2^ Total female sexual dysfunction of 263. ^3^ Odds ratio (OR). ^4^ Adjusted odds ratio (AOR).

## Data Availability

The data presented in this study are available upon request from the corresponding author.

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
