# Peer review of "Impact of the COVID-19 Pandemic on the Female Sexual Function Index and Female Behavioral Changes: A Cross-Sectional Survey Study in Thailand"

_ijerph, 2022, doi:10.3390/ijerph192315565_

Round 1

Reviewer 1 Report

Literature on covid-19 and female sexual behaviour and sexual function/dysfunction not well covered. This ill enhance the discussion.

I recommend you look into the literature. However, I added some references. 

Conclusion can be improved to be more inclusive of what you set out to do in the introduction.

Author Response

For file answer point by point >> Please see the attachment.

Dear Reviewer(s) and Editor.

The article titled "Impact of COVID-19 Pandemic on Female Sexual Function Index and Female Behavioral Changes: A Cross-Sectional Survey Study in Thailand," which was submitted to the IJERPH journal and received the valuable opportunity to be revised by you, has been considered and revised in accordance with your suggestions. Our research team contains two gynecologists who specialize in female pelvic medicine, reconstructive surgery, and sexology, and a scientist who works in the field of epidemiology. We intend to make an article that can provide data for better female healthcare and clinical implications in the future. Thank you for your kindly response and very informative comments to improve the quality of our research. Our research team tries so hard to improve our article to make it the best and get the opportunity to publish in your journal. Moreover, we believe that our article is proper and fits your journal in the special issue, and it will provide interesting data for meta-analysis or cross-country analysis in the future, which can lead to many citations in the future. We are looking forward to getting the good news of our acceptance to publish in your journal.

Thank you for the constructive comments, suggestions, and critiques. We have responded point-by-point below in RED and addressed them in the manuscript using track changes.

Yours sincerely,

Apisith Saraluck, M.D.

Corresponding author: E-mail: apisith.sa@wu.ac.th 

Reviewer 2 Report

·         Title and Abstract:

o   The title is very comprehensive; it lists the outcome variables, the study design and the type of intervention.

o   The abstract is also adequate, it includes the most important aspects of the research but abbreviations should not be used in the abstract (FSFI, RFSD, BMI).

·         Introduction:

o   It should homogenize and properly write Covid-19, throughout the introduction and the manuscript in general.

o   Page 2 line 55: do not express yourself in the first person, it is better to use the impersonal mode. Also, at the end of the introduction (lines 63-65).

o   The sentences: “We discovered several examples in which female sexual dysfunction indicated consultation with obstetrics and gynecology in this condition. Healthcare providers in our area are under greater strain and work longer hours”, should be supported with at least some bibliographical reference.

o   The introduction should end with the objective of the research.

o   Line 60: a dot is left between the words function and and

o   The objective should be: to determine the COVID-19 pandemic effect on female sexual function and to know the risk of female sexual dysfunction.

·         Material and Methods:

o   Page 2, line 68: please delete: non interventional, It´s redundant.

o   How did you contact the study participants?

o   Where does the sample come from?

o   Why are there so many health professionals?

o   How was the sampling?

o   Did you calculate the sample size?

o   A section in the material and methods section should give more details about the procedure carried out to obtain the data.

o   Page 2, line 73: a dot is missing before The Main

o   Was the reliability coefficient of FSFI Thai version?

o   Page 2, line 95 the word "two" is repeated

o   It should specify that Peeyananjarassri et al. only did the translation and back-translation, evaluated comprehension and wording and calculated the reliability coefficient.

o   You should write in full what BMI and CI mean

·         Results:

o   Page3, lines 117 to 119 should be deleted

o   On page 4 (line 133) and 6 (139-141) It is repeated the following sentence:  From the cross-139 sectional study analysis, the incidence of women with presence the risk of female sexual 140 dysfunction (RFSD) during pandemic was 60.8% (263/432). It´s better on page 4.

o   The format of tables 1 and 2 is unfriendly. Please change it.

o   Table 4: Check and complete the headings of the different columns of the table (put percentage, the parenthesis signs properly and remove Ref.)

o   In general, it is not necessary to put all the numerical data in the text since they are already in the tables; for example, in section 3.4.

·         Discussion:

o   The discussion should be deeper and broader. The results of the univariate analysis have not been discussed.

·         Conclusions: Conclusions are redundant.

·         References: Reference 10 is wrong. Instead of Rc R, it should say: Rosen R,

Author Response

(The authors gave the same response as above.)
